# Predicting Outcome of Traumatic Brain Injury: Is Machine Learning the Best Way?

**DOI:** 10.3390/biomedicines10030686

**Published:** 2022-03-16

**Authors:** Roberta Bruschetta, Gennaro Tartarisco, Lucia Francesca Lucca, Elio Leto, Maria Ursino, Paolo Tonin, Giovanni Pioggia, Antonio Cerasa

**Affiliations:** 1Institute for Biomedical Research and Innovation (IRIB), National Research Council of Italy (CNR), 98164 Messina, Italy; roberta.bruschetta@irib.cnr.it (R.B.); giovanni.pioggia@irib.cnr.it (G.P.); 2Department of Engineering, Università Campus Bio-Medico di Roma, Via Alvaro del Portillo 21, 00128 Rome, Italy; 3S’Anna Institute, 88900 Crotone, Italy; l.lucca@istitutosantanna.it (L.F.L.); e.leto@isakr.it (E.L.); m.ursino@isakr.it (M.U.); patonin18@gmail.com (P.T.); 4Pharmacotechnology Documentation and Transfer Unit, Preclinical and Translational Pharmacology, Department of Pharmacy, Health Science and Nutrition, University of Calabria, 87036 Arcavacata di Rende, Italy

**Keywords:** traumatic brain injury, outcome predictors, linear regression, machine learning, ensemble of classifiers

## Abstract

One of the main challenges in traumatic brain injury (TBI) patients is to achieve an early and definite prognosis. Despite the recent development of algorithms based on artificial intelligence for the identification of these prognostic factors relevant for clinical practice, the literature lacks a rigorous comparison among classical regression and machine learning (ML) models. This study aims at providing this comparison on a sample of TBI patients evaluated at baseline (T0), after 3 months from the event (T1), and at discharge (T2). A Classical Linear Regression Model (LM) was compared with independent performances of Support Vector Machine (SVM), k-Nearest Neighbors (k-NN), Naïve Bayes (NB) and Decision Tree (DT) algorithms, together with an ensemble ML approach. The accuracy was similar among LM and ML algorithms on the analyzed sample when two classes of outcome (Positive vs. Negative) approach was used, whereas the NB algorithm showed the worst performance. This study highlights the utility of comparing traditional regression modeling to ML, particularly when using a small number of reliable predictor variables after TBI. The dataset of clinical data used to train ML algorithms will be publicly available to other researchers for future comparisons.

## 1. Introduction

Traumatic brain injury (TBI) has a tremendous impact on patients and family members. They must learn to live with a diminished potential for physical, emotional, cognitive, and social functioning. A recent meta-analysis [1] found an overall incidence rate of 262 per 100,000 per year, and in the USA 43.3% of hospitalized TBI survivors will have a long-term disability [2]. One of the main challenges in TBI-related research is to achieve an early and definite prognosis considering the best predictors of outcome, to administer effective treatments able to improve the clinical progression. Some factors have been proposed and predictive models have been constructed. Most studies used traditional regression techniques to identify these factors [3] defining age, diagnosis, and severity level (measured with Coma Recovery Scale-Revised (CRS-r)) as the most important clinical indicators to predict TBI outcome, although with a poor implementation in clinical practice [4]. In such a scenario, in the last few years, considerable efforts have been put into implementing and developing artificial intelligence tools. Machine learning (ML) methods landed in this neurological domain only a few years ago with promising and enthusiastic perspectives [5]. The first studies tested the performance of support vector machine (SVM) [6], Naïve Bayes (NB) [7], and random forest [8] algorithms in predicting the mortality of TBI patients. Despite good performance having been reported (ranging from 67% to 97%), these preliminary ML studies are characterized by high variability in predictive models and clinical predictors used for the training phase, thus reducing a rigorous comparison among methods. For this reason, in this study, we compare the performance of a Classical Linear Regression Model (LM) with the most common ML algorithms for the prediction of clinical outcomes of TI patients (measured with the Glasgow Outcome Scale-Extension) after 6–9 months from the hospitalization. Finally, we also tested the ensemble ML algorithm and applied different feature selection methods to optimize the model.

## 2. Materials and Methods

### 2.1. Population

The population enrolled for this study was composed of 102 subjects (Table 1). This study was a secondary analysis conducted on a large database used in different previous studies [9,10] and further augmented with new data. For each patient, collected data consisted of demographic information and clinical assessment on admission (T0) after three months (T1) and after 6–9 months at discharge (T2). At the end of the study, 11.8% of TBI patients died, whereas 37.2% had positive outcomes (Glasgow Outcome Scale higher than 4).

### 2.2. Proposed Approach

In this section, we briefly describe the outcome measure, the predictors and the classification methods.

#### 2.2.1. Outcome Measure

As a main measure, we used the extended version of the Glasgow Outcome Scale (GOS-E) [11], which is the most used scale in clinics for outcome assessment after a head injury or non-traumatic acute brain insults. Details about the scales are reported in Table 1. For binary classification, we split the scale into two halves corresponding to Positive and Negative Outcome, respectively. For multi-class classifications, we divided the dataset into four classes, joining together the Lower and Upper sub-categories for the first three classes and PST and Death for the fourth classes as reported in Table 2.

#### 2.2.2. Predictors’ Selection

In the experimental section we have tested the following predictors of outcome:**Age, Sex**.**Marshall classification (T0):** one of the most used systems for grading traumatic brain injury at admission [12]. It is based on the observation of the brain non-contrast CT scan and of the degree of swelling and the presence and size of hemorrhages. Patients are divided into six increasing severity categories: diffuse injury I (no visible pathology), diffuse injury II, diffuse injury III (swelling), diffuse injury IV (shift), evacuated mass lesion V, non-evacuated mass lesion VI. Therefore, the highest categories correspond to the worst prognosis.**Entry Diagnosis (T0):** this categorical variable reports the diagnosis at admission consisting of three possible classes: Vegetative state (VS), Minimally Conscious State (MCS) and Emersion from MCS.**Coma Recovery Scale-Revised (CRS-R**) [13]: the gold standard diagnostic tool for the assessment of DOC patients over the course of recovery. It is composed of six items ranging from 0 to 23, where a higher score corresponds to better functionality.**Rancho Los Amigos Levels of Cognitive Functioning Scale (RLAS)** [14]: scale for the assessment of patients’ cognitive performance. It is composed of eight categories: No Response (Cognitive Level I), Generalized Response (CL II), Localized Response (CL III), Confused—Agitated (CL IV), Confused—Inappropriate—Nonagitated (CL V), Confused—Appropriate (CL VI), Automatic—Appropriate (CL VII), Purposeful—Appropriate (CL VIII).**Disability Rating Scale (DRS)** [15]: scale for the measurement of general functional changes over the course of recovery. It is composed of eight items belonging to four categories: Arousability, Awareness and Responsivity (Eye Opening, Communication Ability, Motor Response), Cognitive Ability for Self-Care Activities (Feeding, Toileting, Grooming), Dependence on Others (Level of Functioning), Psychosocial Adaptability (Employability). The total score ranged from 0 (No Disability) to 29 (Extreme Vegetative State).**Early Rehabilitation Barthel Index (ERBI) A and B** [16]: extended version of the Barthel index for the assessment of early neurological rehabilitation patients over the course of recovery. It contains highly relevant items, such as mechanical ventilation, tracheostomy, or dysphagia and it ranges from −325 to +100.

### 2.3. Feature Selection

In this work, we also applied statistical-based feature selection methods to reduce the computational cost of modeling, improve an easier understanding of data and explore a possible improvement of the performance of the model [17]. According to predictor importance, univariate features ranking was performed using the two most common methods:**Minimum Redundancy Maximum Relevance (MRMR) algorithm** [18]: this method explores the optimal subset of features with the maximum relevance for the response and the minimum redundancy using the pairwise mutual information among features and between each feature and the outcome.**Chi-square Test** [19]: an approach based on individual chi-square tests to examine the relationship between each dependent variable and the outcome.

Next, the optimal subset of features was defined, selecting the highest difference between consecutive scores as a breakpoint and taking the most important predictors. Indeed, a drop among the importance scores represents the confidence of feature selection. Therefore, a large drop indicates that the algorithm is confident in selecting the most important variables, while a small drop suggests that the differences among predictor importance are not significant.

### 2.4. Classification Methods

In the classification phase, we used the LR and the four most conventional classifiers which are described below:**Support Vector Machine (SVM)** [20]: a widely used method based on mapping data into a higher dimensional feature space using kernel functions to make them separable and then finding the best hyperplane for classification. In this study, we used a Radial Basis Function (RBF) kernel:
$$K\left(x_{j},x_{k}\right)=e^{-\frac{\|x_{j}-x_{k}{\|}^{2}}{2\sigma^{2}}}$$
where $x_{j}$ and $x_{k}$ are vectors representing observations *j*-th and *k*-th.

**k-Nearest Neighbors (k-NN)** [21]: a simple approach where each object is assigned to the most common class among its k nearest neighbors applying the majority voting technique. In this study, we set a number of nearest neighbors equal to 5 following a general rule $k=\sqrt{n}$ to identify the optimal value, where *n* is the number of samples in training data [22] and employing the standardized Euclidean distance as a metric:

$$d\left(q,x_{i}\right)=\sqrt{\overset{n}{\underset{i=1}{\sum }}{\left(\frac{q-x_{i}}{\sigma_{i}}\right)}^{2}}$$ where *q* is the query instance, $x_{i}$ is the *i*-th observation of the sample and $\sigma_{i}$ is the standard deviation.

**Naïve Bayes (NB) [23]**: based on Bayes’ Theorem, this technique applies density estimation to the data and assigns an observation to the most probable class assuming that the predictors are conditionally independent, given the class. In this study, probabilities were computed modeling data with Gaussian distribution:


$$f\left(x\right)=\frac{1}{\sqrt{2\pi }}e^{-0.5x^{2}}$$


**Decision Tree (DT)** [17]: based on a tree-like model in which each internal node specifies a test involving an attribute, each branch descending from the node corresponds to one of the possible outcomes of the test and each leaf node represents a class label. Classifying an object with a decision tree means performing a sequence of cascading tests, starting with the root node and finishing with a leaf node. In this study, for the decision tree model, we set a maximal number of decision splits equal to 10. As an algorithm for selecting the best split predictor at each node, we chose standard CART, which selects the split predictor maximizing *Gain*. *Gain* is a split criterion given by the difference between the information needed to classify an object (*I*) and the amount of residual information needed after the value of attribute A have been learned (*Ires*):


$$Gain\left(A\right)=I-I_{res}\left(A\right)$$


where *I* is given by the entropy measure
$$I=-\underset{c}{\sum }p\left(c\right)log_{2}p$$
with *p*(*c*) equal to the proportion of examples of class *c* and
$$I_{res}=-\underset{v}{\sum }p\left(v\right)\underset{c}{\sum }p\left(c|v\right)log_{2}p\left(c|v\right)$$

All these four classifiers were tested with the majority voting ensemble technique. Majority voting is a simple ensemble method that usually is adopted to improve machine learning performances better than any single model used in the ensemble. It works by combining the final classification of all the four ML models (SVM, k-NN, NB and DT). The predictions for each label are summed and the label with the major number of occurrences is the final outcome [24]. ML models were trained and tested using Matlab R2020b (Mathworks, Natick, MA, USA).

### 2.5. Performance Metrics

For the evaluation of the models, we employed two types of stratified Cross-Validation: Leave-One-Out Cross-Validation (LOOCV) and 10-fold Cross-Validation (10-fold CV) [25]. K-fold Cross-Validation is a procedure that consists in splitting the dataset into k subsets and iteratively leaving one subset out as a test set while keeping the remaining subsets together as a training test. Leave-one out is the extreme version of cross-validation where the number of subsets coincides with the number of samples in the dataset. LOOCV requires fitting and evaluating a model for each sample, which maximizes computational cost. The main advantage of this technique is its robustness since, at each iteration, the training set is as similar as possible to real data. This allows unbiased and reliable estimation of performances avoiding overfitting. Ten-fold CV is a commonly used and less computationally expensive version of cross-validation. For applications with real-world datasets, Kohavi recommends stratified 10-fold cross-validation [26]. Classification performances were measured using Accuracy, Precision, Recall and F1-Score [27], defined for multi-classes tasks, as reported in Table 3.

Statistical analysis of performance metrics was carried out using RStudio Version 4.0.3 (10 October 2020) (RStudio, Boston, MA, USA). Since variables were not normally distributed, the Kruskal–Wallis (KW) test was employed to compare performance metrics of ML algorithms to discriminate 2 and 4 classes of the outcome, respectively [28]. The KW test was also used to investigate the performances achieved with different feature selection methods. A *p*-level of <0.05 was used for defining significance, followed by post-hoc Dwass–Steel–Critchlow–Fligner pairwise comparisons.

## 3. Results

No significant differences of performance metrics, respectively, for 2 and 4 classes were found between the MRMR and Chi-square feature selection methods as reported in Table 4 using KW test. Moreover, we observed that MRMR achieved the same performances, respectively for 2 and 4 classes of output as shown in Figure 1 and Figure 2 with a larger drop among the predictor importance scores and selecting a minor number of features useful to reducing computational costs.

Furthermore, we extracted correlations between each pair of clinical predictors as shown in Figure 3 and observed that the variables Marshall, Entry Diagnosis, CRS-R, RLAS and DRS scores were the most correlated features.

For these reasons, we performed further analyses using MRMR with CRS-R, Age and ERBI B for binary classification and Entry Diagnosis, Age and Sex for four-class classification. After feature selection, we applied the KW test between each ML model and LM with a 10-fold CV. Significant differences were found in both cases (Table 5). Post-hoc Dwass–Steel–Critchlow–Fligner pairwise comparisons among accuracies were included to compare each pair of ML models and identify the best performer. Using two classes of outcome we observed a significant difference between LM and NB (see Table 6), although the accuracy of NB is lower than LM (Figure 4). In the case of the four classes of outcome, no significant differences were detected revealing comparable performances among all models, although a significant loss of accuracy was detected (Table 6 and Figure 5). The same trend was observed for other ML metrics. Table 7 and Table 8 reported metric performances using LOOCV and 10-fold CV, respectively, for 2 and 4 classes of the outcome.

## 4. Discussion

In this study, we compare ML approaches to more traditional LM in contemporary TBI patients’ data to predict their clinical evolution, respectively, using 2 and 4 classes of outcome approaches. We demonstrated that classic LM could perform as well as more advanced ML and ensemble ML classifiers in terms of accuracy (sensitivity and specificity) trained by the same predictors.

The LM had the advantage of identifying some prognostic factors, associating each of them with an odds ratio, while the use of ML is limited by the difficulty of interpreting the model, often referred to as ‘black box’. This finding is perfectly in agreement with results recently obtained by Iosa et al. [29], who compared the performance of classical regression, neural network, and cluster analysis in predicting the outcome of patients with stroke. Similarly, Gravesteijn et al. [30] reached the same conclusions on TBI patients evaluating the different performance of logistic regression with respect to SVM, random forests, gradient boosting machines, and artificial neural networks. In terms of model performance, our SVM and DT values are similar to those reported by Abujaber et al. [6], whereas k-NN has never been employed in this clinical domain and this outperformed other ML approaches in all the evaluation metrics. The only algorithm that relatively underperformed was the NB. The accuracy (and sensitivity) was somewhat lower, passing from the analyzed to the test sample. Our data conflict with those reported by Amorim et al. [7] who described the excellent performance of this algorithm as a screening tool in predicting the functional outcome of TBI patients. This discrepancy could be mainly due to the use of different clinical predictors. Indeed, there is large heterogeneity in factors (i.e., age, gender, clinical severity, clinical comorbidities, systolic blood pressure, respiratory rate, lab values, and presence of mass lesion) identified as having a prognostic value in TBI patients, thus making a direct comparison between ML approaches difficult to perform [6]. Another limit is due to the fact the dataset is unbalanced (62.8% negative vs. 37.2% positive outcome) and could negatively affect performances of machine learning. To overcome this issue and increase classification robustness, we also applied the technique of LOOCV that is less affected by this problem and allows us to compare four machine learning techniques since each type of algorithm performs predictions differently. For instance, the DT algorithm performs well with unbalanced datasets thanks to the splitting rules that look at class variables.

Moreover, the type of predictors, such as continuous and categorized (operator-dependent) variables and the lack of objective biological high-dimensional data (i.e., neuroimaging, genetics), might also limit the performance of ML techniques applied in this domain [31]. Our data would seem to confirm this hypothesis because of the change in identified predictors for classification. Indeed, as shown in Figure 4 and Figure 5, moving from 2 to 4 classes of outcome approaches impacts the most significant features extracted by predictive models. Apart from age, for reaching the excellent performance with the 2 classes approach, LM and ML algorithms need CRS-r values and ERBI values at T1, whereas, for the 4 classes approach, diagnosis at admission and sex are the most important features.

Finally, this is the first study employing an ensemble ML approach to improve the outcome prediction in TBI patients. This approach has been demonstrated to be useful for integrating multiple ML models in a single predictive model characterized by higher robustness, reducing the dispersion of predictions [32]. However, this method would not seem to boost performance except when the four classes approach was employed (Figure 5). Indeed, in our KW analysis, we observe that the ensemble ML for two classes reach a high accuracy similar to other ML techniques of about 84% for LOOCV and 82% for 10-fold CV as shown in Table 7 while using four classes approach (Table 8) a (not significant) trend of performance metrics was observed (71.5% for LOOCV and 70.5% for 10-fold CV), which is five to ten percentage points higher than the other models.

## 5. Conclusions

In summary, we found that ML algorithms do not perform better than more traditional regression models in predicting the outcome after TBI. As future work, we plan to perform further external validations of all these models on other datasets that could capture dynamic changes in prognosis during intensive care courses extending the current models with new objective predictors, such as neuroimaging data (EEG, PET, fMRI) [33].

## Figures and Tables

**Figure 1 biomedicines-10-00686-f001:**
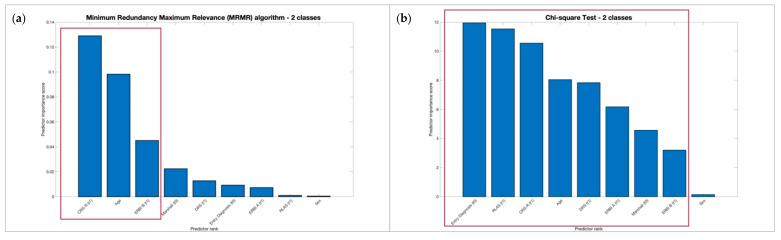
Most significant features selected using 2 classes of outcome, respectively, using MRMR and Chi-Square selection methods.

**Figure 2 biomedicines-10-00686-f002:**
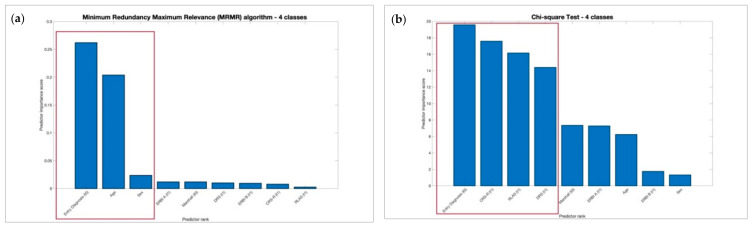
Most significant features selected using 4 classes of outcome, respectively, using MRMR and Chi-Square selection methods.

**Figure 3 biomedicines-10-00686-f003:**
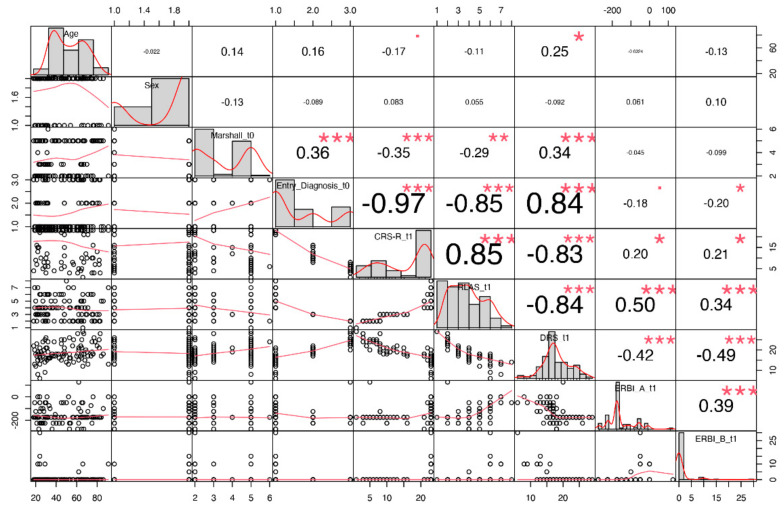
Correlation matrix with paired correlations between each pair of clinical predictors. On the diagonal: distribution of each variable, on the bottom of the diagonal: bivariate scatterplots with a fitted line, on the top of the diagonal: correlation value with significance level. Symbols “***”, “**”, “*”, “.”, indicates respectively *p*-values <0.001, <0.01, <0.05, <0.1.

**Figure 4 biomedicines-10-00686-f004:**
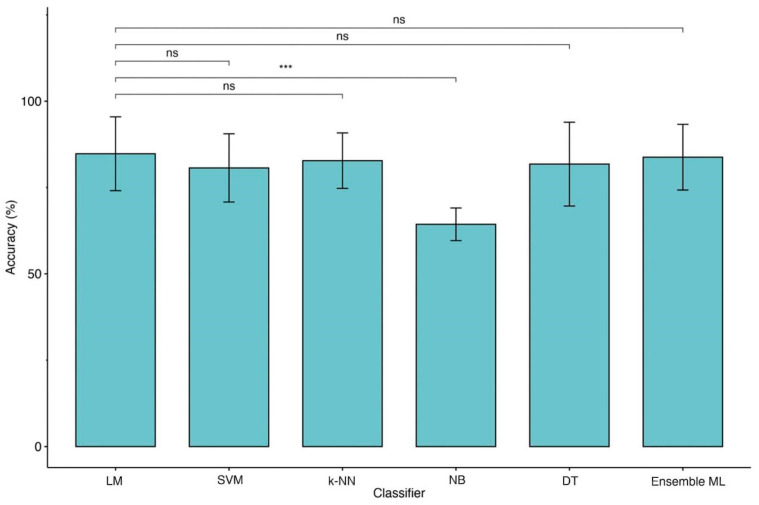
Comparison of accuracies between LM and ML models (2 classes of outcome). Legend: Linear Regression Model (LM), Support Vector Machine (SVM), k-Nearest Neighbors (k-NN), Naïve Bayes (NB), Decision Tree (DT) and Ensemble of Machine Learning models (Ensemble ML). Symbols ***, *p*-values <0.001.

**Figure 5 biomedicines-10-00686-f005:**
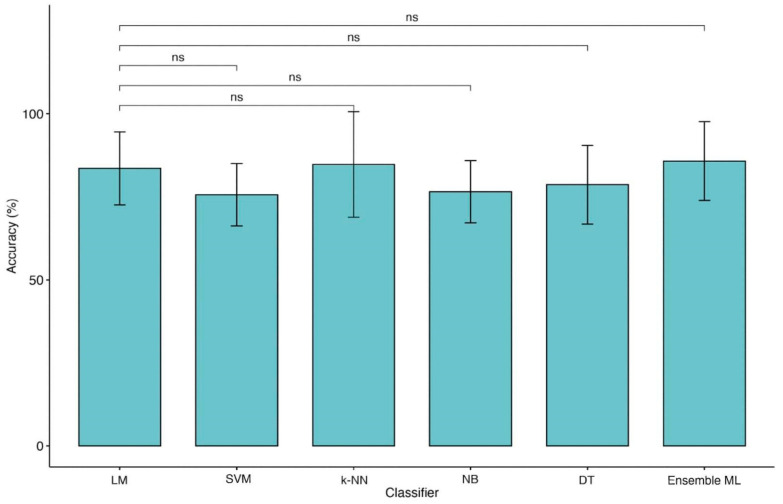
Comparison of accuracies between LM and ML models (4 classes of outcome). Legend: Linear Regression Model (LM), Support Vector Machine (SVM), k-Nearest Neighbors (k-NN), Naïve Bayes (NB), Decision Tree (DT) and Ensemble of Machine Learning models (Ensemble ML).

**Table 1 biomedicines-10-00686-t001:** Clinical characteristics of TBI patients.

Predictors	TBI (n°102)
Age (years)	48 ± 20.6
Sex	29F/73M
Length of stay ICU (days)	27.0 (20–35)
Length of stay IRU (days)	72.0 (43.8–128.2)
Marshall Score ICU (%)	
I	0%
II	38%
III	19.6%
IV	2.2%
V	39.1%
VI	1.1%
Diagnosis at Admission (%)	
Emersion	55.9%
MCS	20.6%
VS	23.5%
Diagnosis at Discharge T2 (%)	
Emersion	86.3%
MCS	8.8%
VS	4.9%
CRS-R T1	23.0 (3–23)
CRS-R T2	23.0 (5–23)
RLAS T1	4 (1–8)
RLAS T2	6 (2–8)
DRS T1	18 (6–29)
DRS T2	9 (0–27)
ERBI T1	−175 (0–275)
ERBI T2	0 (0–175)
Outcome at discharge	
GOS-E values	3.0 (1.0–8.0)
GOS-E range (%)	
negative outcome (<5)	62.8%
positive outcome (>5)	37.2%

ICU: Intensive Care Unit; IRU: Intensive Rehabilitation Unit; MCS, minimally conscious state; VS, vegetative state; GOSE: Extended Glasgow Outcome Scale; CRS-r: Coma Recovery Scale-revised; RLAS: Rancho Los Amigos Scale; ERBI: early rehabilitation Barthel Index; DRS: Disability Rating Scale. Data are presented as median (range).

**Table 2 biomedicines-10-00686-t002:** Glasgow Outcome Scale classes.

Category Number	Name	Definition	4 Classes	2 Classes
**8**	**Good Recovery** Upper	No current problems related to brain injury that affects daily life	**Good Recovery** (26 Subjects)	**Positive Outcome**
**7**	**Good Recovery** Lower	Minor problems that affect daily life;Recovery of the pre-injury level of social and leisure activities > 50%		
**6**	**Moderate Disability** Upper	Reduced work capacity; Recovery of the pre-injury level of social and leisure activities < 50%	**Moderate Disability** (9 Subjects)	
**5**	**Moderate Disability** Lower	Inability to work		
**4**	**Severe Disability** Upper	Autonomy during the day > 8 h; Inability to travel and/or go shopping without assistance	**Severe Disability** (38 Subjects)	**Negative Outcome**
**3**	**Severe Disability** Lower	Necessity of frequent home assistance for most of the time every day		
**2**	**Persistent** **Vegetative** **State**	Unresponsiveness and Speechlessness	**Vegetative/Death** (25 Subjects)	
**1**	**Death**			

**Table 3 biomedicines-10-00686-t003:** Performance Metrics.

Performance Measure	Binary Classification	Multi-Class Classification
**Accuracy**	$\frac{tp+tn}{tp+tn+fp+fn}$	$\frac{\sum_{i=1}^{l}\frac{tp_{i}+tn_{i}}{tp_{i}+tn_{i}+fp_{i}+fn_{i}}}{l}$
**Precision**	$\frac{tp}{tp+fp}$	$P_{\mu }=\frac{\sum_{i=1}^{l}tp_{i}}{\sum_{i=1}^{l}(tp_{i}+fp_{i})}$
		$P_{M}=\frac{\sum_{i=1}^{l}\frac{tp_{i}}{tp_{i}+fp_{i}}}{l}$
**Recall**	$\frac{tp}{tp+fn}$	$R_{\mu }=\frac{\sum_{i=1}^{l}tp_{i}}{\sum_{i=1}^{l}(tp_{i}+fn_{i})}$
		$R_{M}=\frac{\sum_{i=1}^{l}\frac{tp_{i}}{tp_{i}+fn_{i}}}{l}$
**F1-Score**	$2\times \frac{Precision\cdot Recall}{Precision+Recall}$	$F1S_{\mu }=2\times \frac{P_{\mu }\cdot R_{\mu }}{P_{\mu }+R_{\mu }}$
		$F1S_{M}=2\times \frac{P_{M}\cdot R_{M}}{P_{M}+R_{M}}$

In the second column, performance measures defined for binary classification: *tp* represent the true positive, *tn* the true negative, *fp* the false positive and *fn* the false negative. In the third column, the same measures are generalized for a multi-class problem considering *l* classes *Ci*. Macro (*M*) index is the arithmetic mean of all the performance metrics of each class *Ci* and micro (*μ*) index is the global average of each performance metric obtained by first summing the respective *tp*, *fn*, *fp* values across all classes and then implementing the corresponding performance metric equation.

**Table 4 biomedicines-10-00686-t004:** Kruskal–Wallis analysis between MRMR and Chi-square feature selection methods, respectively, using 2 and 4 classes of outcome.

		χ^2^	* p *	ε^2^
** 2 classes **	Accuracy	0.461	0.794	0.0271
	Precision	0.467	0.792	0.0275
	Precision M	1.207	0.547	0.071
	Recall	0.467	0.792	0.0275
	Recall M	0.457	0.796	0.0269
	F1-Score	0.467	0.792	0.0275
	F1-Score M	0.668	0.716	0.0393
** 4 classes **	Accuracy	3.27	0.195	0.1921
	Precision	3.28	0.194	0.1929
	Precision M	1.3	0.523	0.0764
	Recall	3.28	0.194	0.1929
	Recall M	2.54	0.281	0.1493
	F1-Score	3.28	0.194	0.1929
	F1-Score M	2.14	0.343	0.1259

**Table 5 biomedicines-10-00686-t005:** Kruskal–Wallis analysis between Linear model and Machine Learning models, respectively, using 2 and 4 classes of outcome with MRMR algorithm.

		χ^2^	* p *	ε^2^
** 2 classes **	Accuracy	21.6	<0.001	0.367
	Precision	21.6	<0.001	0.367
	Precision M	20.9	<0.001	0.355
	Recall	21.6	<0.001	0.367
	Recall M	26	<0 .001	0.441
	F1-Score	21.6	<0.001	0.367
	F1-Score M	25.3	<0.001	0.429
** 4 classes **	Accuracy	32.8	0.001	0.342
	Precision	26.3	0.001	0.342
	Precision M	20	0.016	0.236
	Recall	26.3	0.001	0.342
	Recall M	20	0.002	0.329
	F1-Score	26.3	0.001	0.342
	F1-Score M	21.2	0.001	0.322

**Table 6 biomedicines-10-00686-t006:** Dwass–Steel–Critchlow–Fligner pairwise comparisons between Linear model and each Machine Learning model using MRMR algorithm.

			Accuracy		Precision μ		Precision M		Recall μ		Recall M		F1-Score μ		F1-Score M	
			W	*p*	W	*p*	W	*p*	W	*p*	W	*p*	W	*p*	W	*p*
**2 classes**	**Linear**	**k-NN**	−0.329	1.000	−0.329	1.000	−0.702	0.996	−0.329	1.000	−0.484	0.999	−0.329	1.000	−0.5367	0.999
	**Linear**	**NB**	−5.116	0.004	−5.116	0.004	−5.400	0.002	−5.116	0.004	−5.714	<0.001	−5.116	0.004	−5.3983	0.002
	**Linear**	**SVM**	−1.315	0.939	−1.315	0.939	−1.184	0.961	−1.315	0.939	−1.665	0.848	−1.315	0.939	−1.6627	0.849
	**Linear**	**DT**	−0.544	0.999	−0.544	0.999	−0.810	0.993	−0.544	0.999	−0.752	0.995	−0.544	0.999	−0.8591	0.991
	**Linear**	**Ensemble**	−0.164	1.000	−0.164	1.000	−0.378	1.000	−0.164	1.000	−0.322	1.000	−0.164	1.000	−0.4299	1.000
**4 classes**	**Linear**	**k-NN**	1.080	0.974	1.080	0.974	0.860	0.991	1.080	0.974	1.180	0.961	1.080	0.974	1.072	0.974
	**Linear**	**NB**	1.414	0.918	1.414	0.918	0.967	0.984	1.414	0.918	1.344	0.933	1.414	0.918	1.341	0.934
	**Linear**	**SVM**	−3.671	0.098	−3.671	0.098	−3.371	0.162	−3.671	0.098	−3.801	0.078	−3.671	0.098	−3.636	0.105
	**Linear**	**DT**	−0.815	0.993	−0.815	0.993	−1.020	0.979	−0.815	0.993	−1.072	0.974	−0.815	0.993	−1.071	0.975
	**Linear**	**Ensemble**	1.680	0.843	1.680	0.843	1.397	0.922	1.680	0.843	1.715	0.831	1.680	0.843	1.768	0.812

**Table 7 biomedicines-10-00686-t007:** Classification Results with MRMR feature selection using, respectively, LOOCV and 10-fold Cross Validation (2 classes).

Classification Model							
Cross Validation	Performance Metric	Linear Model	Support Vector Machine (SVM)	k-Nearest Neighbors (k-NN)	Naïve Bayes (NB)	Decision Tree (DT)	Ensemble
LOOCV	*Accuracy*	84.69%	80.61%	82.65%	64.29%	79.59%	83.67%
	*Precision µ*	84.69%	80.61%	82.65%	64.29%	79.59%	83.67%
	*Precision M*	83.46%	79.51%	81.20%	64.29%	78.14%	82.47%
	*Recall µ*	64.84%	58.09%	61.36%	37.50%	56.52%	63.08%
	*Recall M*	41.51%	38.65%	40.40%	25.00%	38.25%	40.79%
	*F1 Score µ*	73.45%	67.52%	70.43%	47.37%	66.10%	71.93%
	*F1 Score M*	55.44%	52.02%	53.95%	36.00%	51.36%	54.59%
10-fold CV	*Accuracy*	85.89%	81.78%	82.78%	64.33%	76.78%	81.78%
	*Precision µ*	85.71%	81.63%	82.65%	64.29%	76.53%	81.63%
	*Precision M*	84.44%	80.50%	81.20%	64.29%	74.53%	80.00%
	*Recall µ*	66.67%	59.70%	61.36%	37.50%	52.08%	59.70%
	*Recall M*	42.22%	39.37%	40.40%	25.00%	36.75%	40.00%
	*F1 Score µ*	75.00%	68.97%	70.43%	47.37%	61.98%	68.97%
	*F1 Score M*	56.30%	52.87%	53.95%	36.00%	49.22%	53.33%

**Table 8 biomedicines-10-00686-t008:** Classification Results with MRMR feature selection using, respectively, LOOCV and 10-fold Cross Validation (4 classes).

Classification Model							
Cross Validation	Performance Metric	Linear Model	Support Vector Machine (SVM)	k-Nearest Neighbors (k-NN)	Naïve Bayes (NB)	Decision Tree (DT)	Ensemble
LOOCV	*Accuracy*	65.31%	43.88%	66.33%	68.37%	59.18%	71.43%
	*Precision µ*	65.31%	43.88%	66.33%	68.37%	59.18%	71.43%
	*Precision M*	66.49%	36.49%	66.50%	52.94%	46.16%	71.73%
	*Recall µ*	38.55%	20.67%	39.63%	41.88%	32.58%	45.45%
	*Recall M*	36.51%	23.14%	37.28%	38.74%	30.86%	41.58%
	*F1 Score µ*	48.48%	28.10%	49.62%	51.94%	42.03%	55.56%
	*F1 Score M*	47.13%	28.32%	47.77%	44.74%	36.99%	52.64%
10-fold CV	*Accuracy*	64.22%	45.00%	65.33%	67.44%	57.44%	70.44%
	*Precision µ*	64.28%	44.89%	65.31%	67.34%	57.14%	70.41%
	*Precision M*	65.72%	36.83%	65.44%	52.16%	45.83%	70.78%
	*Recall µ*	37.50%	21.36%	38.55%	40.74%	30.77%	44.23%
	*Recall M*	35.73%	23.42%	36.38%	37.78%	28.68%	40.66%
	*F1 Score µ*	47.37%	28.95%	48.48%	50.77%	40.00%	54.33%
	*F1 Score M*	46.29%	28.63%	46.77%	43.82%	35.28%	51.65%

## Data Availability

Datasets collected at S’anna Institute Crotone Italy, all relevant data have been uploaded to a public repository: URL: www.kaggle.com/dataset/6e8c66445ac5b0fda3b3d50cf3a0d1dc4fecb09a3a4e6df19abf98fc0c13a8f3, (accessed on 9 February 2022).

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
