# Peer review of "Predicting Outcome of Traumatic Brain Injury: Is Machine Learning the Best Way?"

_biomedicines, 2022, doi:10.3390/biomedicines10030686_

Round 1

Reviewer 1 Report

The authors of this paper compared the performance of Linear Regression model (LM), Support Vector Machine(SVM), k-Nearest Neighbors(k-NN), Naïve Bayes(NB), Decision Tree(DT), and an ensemble ML approach on the samples of traumatic brain injury(TBI) patients. This study reported the poor performance of NB and some of their findings are consistent with previous publications. However, the authors fail to explain or provide some important rationals and details of their data analysis, which is very important for the evaluation of this paper. For that reason, I would suggest that the following is undertaken to strengthen the existing data and support the conclusions made, before considering the manuscript for potential publication:

Major:

  1. Majority voting is utilized as an ensemble technique in this study, however, the details of the voting system and its performances are missing in the paper. For instance, how would the voting system label a sample? These technical details are needed in this paper;
  2. Feature selection is very important in running ML algorithms. The authors do provide the predictor importance scores in Fig 3 and Fig 4 based on MRMR. However, they did not provide the details of getting these scores and the corresponding scores of Chi-square. These missing parts should be added to the revised version of this paper;
  3. The importance scores for distinct features in 2 classes and 4 classes are very different. For instance, CRS-R is ranking first in the 2 classes but has a very low in the 4 classes. The authors should explain why for the dramatic changes in the same dataset;
  4. Since the ensembled voting system did not outperform other methods, the authors should explain why they thought the ensemble approach is useful?
  5. The dataset in this paper is unbalanced (62.8% negative vs 37.2% positive), the authors should explain if the unbalanced dataset would influence the performances of distinct models.

Minor:

  1. In this paper, Radial Basis Function (RBF) is used as the kernel of SVM. The authors should explain their rationale for choosing RBF other than other kernels;
  2. In k-NN, the k is set to be 5. Would the other value of k influence the performance of k-NN? At least, the authors should explain their rationale for choosing 5 as their k value;
  3. The authors used “Recall ?” and “Recall M” in their results, what is the difference between ? and M?
  4. For the predictor selection, if the relationships or the correlations between each pair of predictors, such as Marshall (t0) and CRS-R (t1), could be provided, it would enhance the quality of this paper;
  5. The formats of the table should be consistent. For instance, the title of Table 4 and Table 5 have a different indent in the current version. The format of these tables should be unified.

Author Response

  1. Majority voting is utilized as an ensemble technique in this study, however, the details of the voting system and its performances are missing in the paper. For instance, how would the voting system label a sample? These technical details are needed in this paper;

REPLY: Thanks for your observation. We improved the description of majority voting into the paper in this way:

“Majority voting is a simple ensemble method that usually is adopted to improve machine learning performances better than any single model used in the ensemble. It works by combining the final classification of all the four ML models (SVM, k-NN, NB and DT). The predictions for each label are summed and the label with the major number of occurrences is the final outcome [23].”

2. Feature selection is very important in running ML algorithms. The authors do provide the predictor importance scores in Fig 3 and Fig 4 based on MRMR. However, they did not provide the details of getting these scores and the corresponding scores of Chi-square. These missing parts should be added to the revised version of this paper;

REPLY: Following the reviewer’s suggestion we improved the description of predictor importance scores in the first part of the Results section in this way: “No significant differences of performance metrics respectively for 2 and 4 classes emerged between the MRMR and Chi-square feature selection methods as reported in Table 4 using KW test. Moreover, we observed that MRMR achieved the same performances respectively for 2 and 4 classes of output as shown in Figure 3 and 4 with a larger drop among the predictor importance scores and selecting a minor number of features useful to reduce computational costs. ”

We also added plots with importance scores of the Chi-square Test respectively in Figures 3 and 4 as suggested.

3. The importance scores for distinct features in 2 classes and 4 classes are very different. For instance, CRS-R is ranking first in the 2 classes but has a very low in the 4 classes. The authors should explain why for the dramatic changes in the same dataset;

REPLY: Feature selection is one of the crucial steps of machine learning. It is very complex to identify the most relevant and informative variables even because their combination could modify their predictive power. The reason why a variable could be very useful in classifying two groups and lose its discriminative power splitting the dataset into more classes is attributable to data distribution. In the discussion, we now included a new statement about the importance of data distribution in this domain.

4. Since the ensembled voting system did not outperform other methods, the authors should explain why they thought the ensemble approach is useful?

REPLY: As reported in paragraph 4 “Discussion”, the ensemble approach has been demonstrated to be useful for integrating multiple ML models in a single predictive model characterized by higher robustness, reducing the dispersion of predictions. However, in our application, as confirmed by Kruskal-Wallis analyses, the ensemble voting system achieves not significant outcomes for 2 classes (Table 7), we observe that in Table 8 (4 classes output) there is a trend of ensemble model performance metrics some percentage points higher than the other models. We improved this part into the Discussion session in this way: “Indeed, in our KW analysis we observe that the ensemble ML for two classes reach a high accuracy similar to other ML techniques of about 84 % for LOOCV and 82% for 10-fold CV as shown in Table 7, while using four classes approach (Table 8) a (not significant) trend of performance metrics was observed (71,5% for LOOCV and 70,5 % for 10-fold CV), which is five to ten percentage points higher than the other models.”

5. The dataset in this paper is unbalanced (62.8% negative vs 37.2% positive), the authors should explain if the unbalanced dataset would influence the performances of distinct models.

REPLY: We improved this part into the Discussion session in this way: “Another limit is due to the fact the dataset is unbalanced (62.8% negative vs 37.2% positive outcome) and could negatively affect performances of machine learning. To overcome this issue and increase classification robustness, we also applied the technique of LOOCV that is less affected by this problem and allows us to compare four machine learning techniques since each type of algorithm performs predictions differently. For instance, the DT algorithm performs well with unbalanced datasets thanks to the splitting rules that look at class variables.”

Minor:

  1. In this paper, Radial Basis Function (RBF) is used as the kernel of SVM. The authors should explain their rationale for choosing RBF other than other kernels;

REPLY: Kernel selection for Support Vector Machine, is strictly related to the specific problem to solve. Among many possibilities, the most used are the linear kernel, radial basis function (Gaussian) kernel and polynomial kernels. The linear kernel is the basic type and it is usually employed for data that are linearly separable, in fact, it is largely used in text classification. Moreover, applying a linear kernel is equivalent to logistic regression. Here we sought to compare different and more complex machine learning approaches with respect to the classical linear regression model. Polynomial kernels are more general and mainly employed in image classification. In addition, they appear less accurate and efficient. Gaussian radial basis function instead is the most used for non-linear data because it's localized and has a finite response along the complete x-axis. [S. Pahwa and D. Sinwar, “Comparison Of Various Kernels Of Support Vector Machine,” Int. J. Res. Appl. Sci. Eng. Technol., vol. 3, no. VII, pp. 532–536, 2015].

2. In k-NN, the k is set to be 5. Would the other value of k influence the performance of k-NN? At least, the authors should explain their rationale for choosing 5 as their k value;

REPLY: The employment of the optimal k value is fundamental but it might be tricky. A small value might increase noise with a higher influence on the result whereas a large value increases computational expensiveness. If there is an even number of classes, odd values of k are preferred in order to avoid doubts in classification. Moreover, the general rule to identify the optimal value is defined by the formula now included in the methods section, where n is the number of samples in training data. We employed this rule to find the order of magnitude but next, we also performed a parameter tuning choosing 5 as the optimal value to achieve the best performance with less computational effort. We improved this part into the “Classification methods” section.

3. The authors used “Recall ?” and “Recall M” in their results, what is the difference between ? and M?

REPLY: We explained the concept of ? and M into the description of Table 3 integrating the following text: “Macro (M) index is the arithmetic mean of all the performance metrics of each class Ci and micro (μ) index is the global average of each performance metric obtained by first sum the respective tp, fn, fp values across all classes and then implementing the corresponding performance metric equation.”

  1. For the predictor selection, if the relationships or the correlations between each pair of predictors, such as Marshall (t0) and CRS-R (t1), could be provided, it would enhance the quality of this paper.

REPLY: We added and explained Figure 3 into the Results section.

  1. The formats of the table should be consistent. For instance, the title of Table 4 and Table 5 have a different indent in the current version. The format of these tables should be unified.

REPLY: Done

Reviewer 2 Report

The manuscript “Predicting Outcome of Traumatic Brain Injury: is machine learning the best way?” reports an interesting study on comparison among classical regression and machine learning models of traumatic brain injury. Artificial intelligence and machine learning are landed in neurological domain only a few years ago with promising and enthusiastic perspective. Although, this manuscript reports data of ‘a secondary analysis conducted on a large database’, it is interesting for ‘artificial intelligence and machine learning’ group.  

  1. There is discrepancy between T1 group as ‘it is mentioned as after 4 months in abstract’, and ‘after 3 months in materials and methods section’.
  2. It is interesting to know that ‘No significant differences emerged among feature selection methods’; however, authors should explain their ‘selection of the KW test between each ML model and LM with 10-fold CV’. Why is it not interesting to apply KW test between each ML model and LOOCV?
  3. This is fair to know that applied ML algorithms do not perform better than more traditional regression models in predicting the outcome after TBI. However, authors should be more specific on discussion with limitations of the models and how can be improved.

Author Response

  1. There is discrepancy between T1 group as ‘it is mentioned as after 4 months in abstract’, and ‘after 3 months in materials and methods section’.

REPLY: Modified

2. It is interesting to know that ‘No significant differences emerged among feature selection methods’; however, authors should explain their ‘selection of the KW test between each ML model and LM with 10-fold CV’. Why is it not interesting to apply KW test between each ML model and LOOCV?

REPLY: As reported in paragraph 2.5 “Performance Metrics”, LOOCV is the extreme version of the cross-validation technique where the number of subsets coincides with the number of samples in the dataset. LOOCV requires fitting and evaluating a model for each sample. So, the outcome of each iteration is referred to as a single sample. As a consequence, performance metrics can only assume two possible values: 100% or 0%, differently from 10-fold CV where, at each iteration, the mean among all the samples of the test set is computed. Dichotomous variables are not suitable to perform a statistical comparison among several groups where the normal distribution of data is required. Moreover, as we can see from tables 7 and 8, results obtained with the two methods of cross-validation are very similar, confirming the stability and reliability of classification.

3. This is fair to know that applied ML algorithms do not perform better than more traditional regression models in predicting the outcome after TBI. However, authors should be more specific on discussion with limitations of the models and how can be improved.

REPLY: We would like to thank this reviewer for this suggestion. We now improved the  Discussion section as requested.

Round 2

Reviewer 1 Report

All my concerns have been addressed

Reviewer 2 Report

The manuscript is revised well, and authors responded satisfactorily the reviewers' comments. The revised version is recommended for publication.